# Effectiveness of an Intervention to Enhance First Aid Knowledge among Early Childhood Education Students: A Pilot Study

**DOI:** 10.3390/children10071252

**Published:** 2023-07-21

**Authors:** Patxi León-Guereño, Laura Cid-Aldama, Héctor Galindo-Domínguez, Alaitz Amezua-Urrutia

**Affiliations:** 1Department of Physical Activity and Sport Science, Faculty of Education and Sport, University of Deusto, 20012 Donostia-San Sebastián, Spain; 2Department of Didactics and School Organization, Faculty of Education and Sports, University of the Basque Country, 01006 Vitoria-Gasteiz, Spain; lcid007@ikasle.ehu.eus (L.C.-A.); hector.galindo@ehu.eus (H.G.-D.); 3Department of Education, Faculty of Education and Sport, University of Deusto, 20012 Donostia-San Sebastián, Spain; alaitz.amezua@deusto.es

**Keywords:** first aid, early childhood, education, intervention, knowledge

## Abstract

Empowering early childhood education students from the beginning with the necessary knowledge and skills to act swiftly in emergency situations could be crucial in saving lives in certain cases. In order to improve the first aid knowledge and skills of early childhood education students, a pre/post study was conducted with a two-week intermediate intervention involving 13 early childhood education students. Their knowledge and skills in first aid were assessed using an ad-hoc instrument before and after the intervention. The results demonstrate a statistically significant improvement in all items related to first aid general knowledge, first aid kits, and CPR maneuvers, as well as in the overall scale. These findings provide evidence that early childhood education students can be equipped through low-cost interventions to acquire and apply certain essential first aid skills, such as dialing emergency services, understanding the purpose of first aid kit items, and recognizing vital signs in individuals, that may be crucial in saving lives in the future.

## 1. Introduction

First aid involves promptly providing care to an injured or sick individual, typically administered by a non-professional and within a limited range of skills. It is usually carried out until the injury or illness is effectively addressed, such as in the case of minor cuts, bruises, or blisters, or until further medical assistance, such as an ambulance or doctor, becomes available. From a public health standpoint, it is always preferable to prevent injuries or illnesses rather than having to treat those affected [1].

Significant efforts have been made in the educational field in recent years to understand the prevalence of different actions for using first aid [2]; however, most of the investigations have focused on teachers’ knowledge [3,4] and in primary or secondary educational contexts, while few have been developed for early childhood or kindergarten students [2,5,6,7]. Despite the fact that there may be some actions that can be taught to early childhood education students, like cuts, choking, burns, intoxication, hemorrhages, fractures, fainting, venomous animals bite, or emergency calls [8], the more appropriate educational methods and their effectiveness according to children’s age ranges, as well as best teacher methodologies and characteristics of trainers, are still to be properly defined [9]. Moreover, the interventions carried out in the school context overall differ among different investigations [10], thus not being clear which type of intervention is more effective [9]. Within the different types of first aid training programs in the formal educational context or school curricula, the most common characteristics are highlighted in the meta-analysis carried out by De Buck et al. in 2015 [5], mainly being these four actions: enhancing helping behaviors, learning how to call for emergencies, recognizing choking, and moving individuals into recovery positions.

First, there is some evidence of the prevalence of helping behavior among children. Indeed, in children between the ages of 11 and 19, the willingness to provide basic life support ranges from 27% to 38% [11,12], while the willingness to perform chest compressions ranges from 31% to 55% [12,13]. According to Hubble et al., about 43% of children are willing to provide mouth-to-mouth ventilation [12]. A study found that the readiness to perform CPR was not associated with theory scores or independent skills assessment [14], although two other studies demonstrated that training would increase significantly the willingness to help and perform CPR [12,15]. Confidence in providing assistance was also measured in different studies. After training, more than half of the children involved in the study aged 11–12 believed they would be capable of saving a life [14], and the self-efficacy of children aged 13–14 was higher significantly [15]. Several aspects negatively influenced behaviors towards helping, including the fear of failure [12], fear of causing harm to the victim [12,13], fear related to disease transmission [11,12,13,14], encountering a dirty casualty, bleeding [11,13,14], dangers to the rescuer, assisting a stranger [11,12,13,14,16], and the fear of legal repercussions [13]. On the other hand, factors that positively influence attitudes and behavior toward helping include assisting a relative [12,13,14,16] or another child [12,13,14] and having previous first aid training [12].

Second, different studies were conducted to assess the knowledge and skills related to calling the emergency number, this being one of the most used contents in previous intervention research [9]. In a study involving 5- and 6-year-old children, significant differences in knowledge and skills were not found among those who received training and those without training [17]. However, in 2 studies with children aged 6–7 and 10–12, after training, a significant improvement in knowledge of the emergency number was shown. For example, in the 6–7-year-old group, the knowledge increased from 16% to 77% [18,19]. On the other hand, 1 study involving 11–12-year-old children did not show such improvement [20]. The 6–7-year-old children also demonstrated improved skills, providing correct emergency call information compared to the untrained group [18]. In another study, some primary education students showed a significant improvement in calling the emergency number from 35% to 69% 1 week after training [21]. However, the ability to provide necessary information did not show a significant improvement in 13–16-year-olds who underwent CPR training compared to those without training [22]. The ability to call the emergency number was scenario-dependent for children aged 11–16 one year after first aid training [23]. Overall, it can be concluded that children between 6 and 16 years of age are capable of calling the emergency number.

Third, with regard to choking, previous research showed that training significantly improved knowledge of first aid for choking in primary and secondary children [19,24]. With regard to recovery position, some studies demonstrated that training resulted in significant improvements in knowledge and skills related to the recovery position in primary education students [[18],[20],[23]，[25],[26],[27]] even though more research is needed, especially when it comes to properly addressing early childhood education for both choking and the recovery position and first aid-related general knowledge [9].

With regard to primary education interventions, Bánfai et al. [28] examined a group of children who received first aid training that showed a significant improvement regarding their knowledge of first aid techniques. Bollig et al. [18] analyzed the impact of a first aid training program on the performance of school children in a first aid scenario, and the results showed a significant improvement in trained students. Similarly, Lubrano et al. [29], in the Italian context, assessed the benefits of teaching emergency procedures, showing improvements and the importance of knowledge retention and drew upon information from an American Heart Association manual for basic life support [30]. Connolly et al. [20] in Northern Ireland carried out a four-stage stepped CPR training, and the results indicated a significant increase in first aid knowledge after the training. More recently, Soni and Soni [31] revealed in a sample of early adolescents how after an intervention, knowledge regarding first aid and safety measures was improved significantly.

With regard to early childhood education interventions, Plischewski et al. [32] conducted an intervention called Henry with 50 preschool students aged 4 to 6 years and 588 preschool teachers. The findings of this study revealed that the preschool students, after participating in the program, improved their knowledge of how to respond to burns, choking, and unconsciousness. Likewise, Bollig et al. [33] carried out a pilot study with a small sample of kindergarten 4–5 years old children. This study used a mixed method combining qualitative and quantitative data collection. First aid training was provided followed by an assessment of the children in the first aid scenario two months after the training. Subsequently, continuous observation of the participants was carried out over several months in contrast to the official test scenario with the aim of analyzing the children’s knowledge of the game situation, application of first aid in everyday life, and demonstration of skills. The findings suggested that kindergarten children can learn and apply basic first aid, leading to the recommendation and suitability of first aid training at this age. By the same token, in a study conducted by Ammirati et al. [34] involving French kindergartners, the children were shown pictures of different accident situations and asked how they would help the people depicted. Some images showed real injuries (e.g., a child lying near a ladder), while others did not (e.g., a girl crying because her doll was damaged). The children’s responses were evaluated according to a predefined framework; thus, after receiving basic first aid instruction from their teachers, the children’s responses were assessed using a predefined framework, which indicated an improvement in their comprehension of first aid situations. Another first aid training investigation carried out with 150 kindergarten children showed that this type of training course helps preschool children to act and provide help in emergency situations [35]. In a recent pilot study carried out with kindergarten students, pupils’ knowledge and skills were significantly improved regarding first aid, showing some preliminary results that indicated that first aid training may be beneficial for preschool students [7]. These studies provide some evidence suggesting that young children can be effectively taught first aid skills, although the long-term retention of these skills remains uncertain. The majority of research in this area has centered around European children between the ages of 4 and 12, with first aid training being delivered by either teachers or trained professionals [34].

According to Lenson and Mills’ meta-analysis [36], it was found that there is insufficient reliable evidence to effectively determine the best training methods and enhance the retention of first aid general knowledge among school children, and this gap gets bigger when it comes to kindergarten pupils [7] and still bigger when it comes to the youngest pupils within kindergarten since three-year-old pupils have barely been addressed. Therefore, it can be said that more research is needed in this field [9]. By formally evaluating first aid training, we can establish guidelines for training and intervention methods and first aid knowledge in school children. This in turn will contribute to strengthening the community’s ability to provide effective first aid.

Based on this gap, the present study attempts to shed some light on the effectiveness of teaching preschool children first aid. For this purpose, a two-week intervention was designed and carried out by the teacher and part of the research team in order to see how the planned three activities would influence the analyzed three-year-old kindergarten pupils. Therefore, the aim of this study was to see whether it is possible to increase first aid knowledge and skills among a sample of preschool students through a low-cost intervention.

## 2. Materials and Methods

### 2.1. Sample

This research was conducted in a public school located in a small town with almost four thousand inhabitants. Taking this into account, this school is the only one in the village, and its educational offer is for students two years old to twelve years old, i.e., infant and primary education. As far as methodology is concerned, they work with learning situations; in other words, they have a project-based methodology. In addition, in the infant education stage, open circulation and free play are practiced, always giving them tools and designing the environment to the learning situation at the time. The research was carried out specifically in the classroom of 3-year-old children in early childhood education. In this classroom, there were 18 students, 10 girls, and 8 boys, and the unique inclusion criteria taken into account was the fact that the student must be enrolled in the class where the experience was carried out It should be noted that overall attendance in this school is very irregular. Due to this reason, the research was carried out with 13 students (9 girls and 4 boys). Out of these 13 students, 2 had selective mutism issues. Therefore, it was challenging to implement the method with them. Additionally, a child with difficulties in understanding is also included in the research group. Dealing with these factors has made the implementation of the method complex. In terms of age, all the children are three years old, but each child’s abilities differ despite their same age.

### 2.2. Instruments

An ad-hoc questionnaire, displayed in Table 1, was used for this research. This questionnaire includes ten questions related to initial knowledge about first aid. For designing this questionnaire, previous research that highlights the knowledge children should have about first aid was taken into account [37]. A three-point scale was used to analyze the response to each question.

The scale values are as follows: Knows (2 points), Partially knows (1 point), and Doesn’t know anything (0 points). The aim of this scale is to assess the level of knowledge that children have on the subject.

The initial questionnaire was conducted before the interventions, and the final questionnaire was conducted after the interventions with a gap of approximately two weeks. This way, it could truly be observed how the knowledge was consolidated and whether it was forgotten over time.

### 2.3. Procedure

The participating center was a preschool located near the research team. Prior to the experimental work, the intervention was jointly designed by the classroom teacher and part of the research team. During this process, informed consent was obtained from both the school’s management team and the students’ families. Once the intervention was designed, the pre-questionnaire phase was conducted during regular school hours. A few days later, the intervention took place, and shortly thereafter, the data from the post-phase were collected. Due to the young age of the students, the questionnaire data were answered by the research team through direct contact and conversation with the students. Throughout the process, the anonymity and privacy of the students were safeguarded, as no personal information, such as home address, was requested from them. The data were stored in a database hosted in a repository that only the research team has access to.

### 2.4. Data Analysis

To compare the results of the initial and final questionnaires, SPSS Statistics 25.0 was used. Indeed, to observe significant differences between the means of the two groups (pre and post), a *T*-test was used. Additionally, Cohen’s d was used to determine the effect size between the two groups. These results facilitated the evaluation of the intervention and reflection on the different items.

### 2.5. Intervention

The intervention consisted of implementing three different activities by the teacher, each with the intention of addressing a specific objective (Figure 1). The first activity involved holding a discussion about what first aid is, what the number 112 is, and its purpose. Subsequently, an educational video was shown, emphasizing the usefulness of the number 112 in emergency situations. The activity concluded with creating a poster using pictograms that highlighted the purpose of the number 112. This activity lasted for 20 min.

The second activity aimed to familiarize the students with what a first aid kit is and the purpose of the various items in it. Initially, a small discussion was held with the students to debate their understanding of a first aid kit and its purpose. Then, the students participated in creating a first aid kit, and the teacher explained the purpose of each item using pictograms once again. This activity lasted for 20 min.

Finally, the third activity focused on identifying breathing and heart rate. The students used a stethoscope to listen to their heartbeats. In pairs, while one student lay down, the other observed the movements of their belly and felt for breath under their nose, both clear indicators of breathing. This activity lasted for 15 min.

## 3. Results

Initially, regarding the questions related to Activity 1 and as collected in Table 2, a significant rise in general first aid knowledge was observed among preschool students following the intervention. Specifically, it was observed how the level of knowledge regarding what first aid is improved, as well as the understanding of the emergency phone number. Similarly, the students enhanced their knowledge by internalizing the P.N.A. protocol, as well as by learning how to protect themselves to a greater extent, identify accidents, and understand the steps to take in case of an accident.

Regarding the questions related to Activity 2, as depicted in Table 3, it was observed that preschool students increased their knowledge about the essential items that a first aid kit should include, as well as their understanding of the purpose of each instrument.

Finally, regarding the questions from Activity 3, as shown in Table 4, it was observed that preschool students increased their knowledge about CPR maneuvers, as well as improved their ability to identify vital signs, such as pulse and respiration.

After calculating all the statistics for the different items, it was evident that the students achieved a statistically significant improvement with a high effect size (*p* < 0.001; d = 1.04) in the assessed knowledge and skills from the pre-phase (M = 4.31; SD = 3.63) to the post-phase (M = 9.92; SD = 6.68).

## 4. Discussion

The purpose of this study was to determine whether it is possible to increase first aid knowledge and skills among a sample of preschool students through a low-cost intervention. These results revealed that the students who participated in the intervention improved significantly their first aid knowledge and skills. This finding is consistent with previous evidence carried out with early childhood students [33,34,35] and with more recent investigations with kindergarten students that showed that first aid general knowledge and skills were significantly increased after a training program [7,32]. Likewise, our results are in consonance with previous research that showed that training programs were significantly increasing first aid general knowledge among older students [11,12,15,29], thus showing the efficacy of this type of training sessions. Among this investigation, the research carried out by Mohajervatan et al. [35] deserves special attention since it was carried out with a 150-kindergarten sample and showed improvements in children’s first aid knowledge, with these results being in consonance with our results in our investigation.

Within students’ first aid general knowledge and skills related to calling emergency number 112, our results are in line with previous investigations that showed that kindergarten students’ knowledge was significantly improved after training [18,19] and more recently with an investigation carried out with 4–5 years old pupils [7] in which through a first aid training, kindergarten pupils showed a significant higher knowledge. Likewise, some previous research carried out with an 11–12-year-old sample showed a significant improvement in calling the emergency number [21]. On the other hand, our results are not supported by some previous studies involving 5- and 6-year-old children [17] and 10–12-year-old pupils [20] since these investigations did not show significant differences in skills and knowledge among those who received training and those without training, thus not being consistent with our results.

Regarding the CPR maneuver, our results showed that the intervention carried out with our three-year-old pupil group showed a significant improvement. Along the same line, previous interventions carried out with school children showed a significant improvement in CPR knowledge and a willingness to help [12,13,15,16,20] even though these cited investigations were carried out mostly in the high school context with many less analyzed, younger pupils. In this line, within the mixed contents that different first aid interventions show in [9] previous research, in line with our investigation, the use of resuscitative and non-resuscitative programs can provide kindergarten pupils with first aid training [7,32,35] is an aspect that needs to be considered.

However, some previous research did not show such significant improvement in the willingness to help the pupils [14], and the ability to provide necessary information did not show a significant improvement in 13–16-year-olds who underwent CPR training compared to those without training [22]. Therefore, it seems necessary to keep on the research in this field in order to better address the effect of training programs on the improvement of first aid knowledge.

Even though in the same line as our results, recent investigations [1,7] found that children may benefit from first aid training., they concluded that further studies were needed in order to determine the first aid role in kindergarten. Moreover, two potential solutions that could be considered to tackle this challenge include promoting a first aid culture within society and broadening the accessibility of first aid education. Currently, first aid education is predominantly delivered through private institutions, which can pose financial barriers for individuals unable to afford it. Research indicates that schools can play a beneficial role in advancing this objective [34], and therefore the inclusion of this type of training should be seriously considered.

Our findings are relevant to preschool teachers, as they show that it is possible to increase first aid knowledge from the early educational stages, as a lack of training in teachers has been detected [3,38]. This may interest them in preparing a specialized curriculum with simple and practical activities on teaching children how to respond in case of needing help [12,13], making emergency calls [18,19,21], handling choking incidents [19,24], or moving someone into the recovery position [18,20,25,26].

Finally, this study is not without limitations that should be taken into account. Firstly, the sample size collected is small due to the complexity of data collection with such young participants [7]. Therefore, future studies should replicate the same study design, aiming to expand the sample size of the present study. Secondly, the intervention only assessed the students’ knowledge before and after the intervention without evaluating the retention of knowledge months after the intervention as some previous studies have done [18]. Considering the loss of knowledge that students may experience in this subject, it could be relevant to periodically plan small reinforcement training sessions for preschool students. Finally, thirdly, the intervention was evaluated using a non-psychometrically validated ad-hoc instrument assessed by the research team. Therefore, it may be interesting for future work to design a validated instrument to assess the first aid skills and knowledge of preschool students that can be evaluated from different perspectives, such as teachers or families. In spite of these limitations, it is expected that this study contributes to expanding the existing knowledge of first aid among preschool students and serves as an important resource for future research in this area.

## 5. Conclusions

Based on our findings, the three-year-old pupil group taking part in this research achieved a statistically significant improvement in the assessed first aid-related knowledge and skills, showing a high effect size from pre-phase to post-phase, thus filling a described limitation in the literature [9]. Our findings could be relevant in the education system and from a public health perspective. Similarly, when it comes to practitioners, this research also shows the need for initial teacher training programs for preschool educators to empower them with the basic knowledge required by their future students [4]. This would enable them to carry out effective low-cost interventions as well. Likewise, this data is relevant to lawmakers and politicians, as it highlights the importance of designing a robust first aid curriculum for the preschool stage [5]. Therefore, it can be said that first aid training should begin at this pre-school stage [33], thus forming and ensuring knowledge and skills that will advocate for public health. By formally evaluating first aid training, we can establish guidelines for training and intervention methods and first aid knowledge in school children. This in turn will contribute to strengthening the community’s ability to provide effective first aid. Regarding future research, the literature in this field shows a clear limitation; the analyzed groups are small [9], so researchers should work on trying to fill this gap [4].

## Figures and Tables

**Figure 1 children-10-01252-f001:**
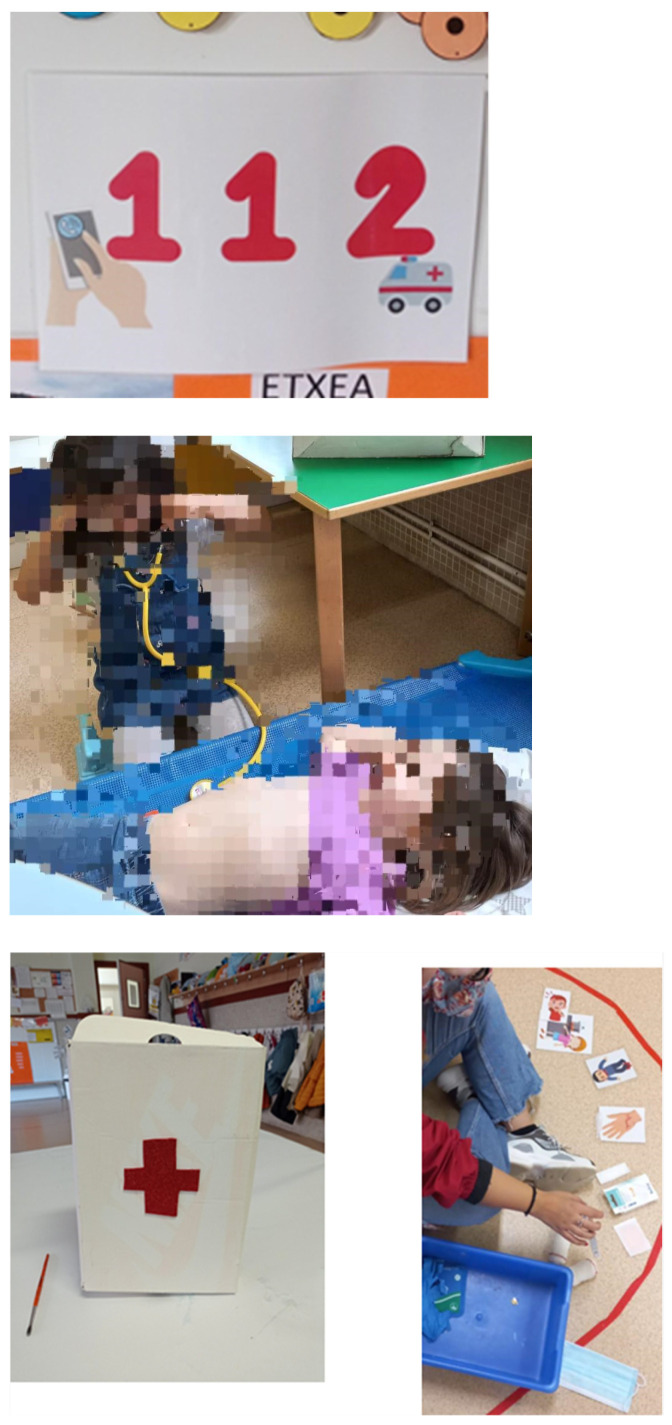
Images from the activities.

**Table 1 children-10-01252-t001:** Instrument used to measure the level of knowledge on first aid.

	Known (2)	Partially Known (1)	Unknown (0)
Have you heard the word “first aid” before?	Knows clearly what they are and why they are used.	Knows that assistance should be provided but doesn’t know what “first aid” means.	Doesn’t have any idea what “first aid” means and why they are used.
Can you identify which types of accidents could happen?	Identifies more than 4 different types of accidents.	Identifies 2 types of accidents.	Doesn’t identify any types of accidents.
Do you know what to do in front of an accident?	Knows the P.N.A. protocol.	Knows that someone should be notified.	Doesn’t know what to do in such a situation.
Do you know what the letters P.N.A. (Protect, Notify, Assist) mean?	Identifies all the abbreviations correctly.	Only identifies one abbreviation.	Doesn’t identify any abbreviations.
Do you know how to protect yourself in front of a risk?	Knows what one should do to protect themselves.	Only knows how to protect themselves in certain situations.	Doesn’t know how to protect themselves.
Do you know the emergency number 112?	Identifies the number 112 and knows what it is used for.	Recognizes the number 112 but doesn’t know its purpose.	112 is completely unknown to them.
Can you identify vital signs? Can you identify pulse and breath?	Identifies both correctly.	Only identifies one of them.	Doesn’t know how to identify vital signs.
Do you know what should be in a first aid kit for initial care?	Knows more than 4 elements.	Knows 2 elements.	Doesn’t know any elements.
Do you know what items are in a first aid kit and what they are used for?	Knows the use of 4 elements.	Knows the use of 2 elements.	Doesn’t know the use of any elements.
Do you know what the CPR maneuver is?	Knows the entire maneuver.	Knows the maneuver but doesn’t know all the steps.	Has no idea what the maneuver is.

CPR: Cardiopulmonary resuscitation; P.N.A.: Protect, Notify, Assist.

**Table 2 children-10-01252-t002:** Questions related to first aid general knowledge Activity 1.

	PRE-Phase		POST-Phase		*p*	d
	M	SD	M	SD		
Have you heard the word “first aid” before?	0.230	0.439	1.15	0.899	<0.001	1.30
Do you know the emergency number 112?	0.380	0.506	1.46	0.877	<0.001	1.50
Do you know what the letters P.N.A. (“Protect, Notify, and Assist”)?	0	0	0.540	0.519	0.001	−1.47
Do you know how to protect yourself in front of a risk?	0.310	0.480	0.620	0.650	0.020	0.542
Can you identify which types of accidents could happen?	0.850	0.801	1.08	0.862	0.041	0.276
Do you know what to do in front of an accident?	0.380	0.506	0.850	0.801	0.004	0.701

M, Mean; SD, Standard Deviation; *p*, *p*-value; d, Cohen’s d for measuring effect size; P.N.A.: Protect, Notify, Assist.

**Table 3 children-10-01252-t003:** Questions related to the items that first aid kit should include and the use of each item, Activity 2.

	PRE-Phase		POST-Phase		*p*	d
	M	SD	M	SD		
Do you know what should be in a first aid kit for initial care?	0.920	0.641	1.46	0.660	0.001	0.830
Do you know what items are in a first aid kit and what they are used for?	0.920	0.641	1.15	0.801	0.041	0.317

**Table 4 children-10-01252-t004:** Questions related to PCR maneuver knowledge and vital signs related, Activity 3.

	PRE-Phase		POST-Phase		*p*	d
	M	SD	M	SD		
Do you know what the CPR maneuver is?	0	0	0.380	0.506	0.009	1.06
Can you identify vital signs? Can you identify pulse and breath?	0.310	0.480	1.23	0.832	<0.001	1.20

CPR: Cardiopulmonary resuscitation.

## Data Availability

The data that support the findings of this study are available from the corresponding author upon reasonable request.

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
