# Peer review of "Effectiveness of an Intervention to Enhance First Aid Knowledge among Early Childhood Education Students: A Pilot Study"

_children, 2023, doi:10.3390/children10071252_

Round 1

Reviewer 1 Report

Dear authors

I received your submission as a reviewer and have read it several times with great interest. I appreciate your great job which was so exciting in terms of conduction. However, there are several important concerns from my viewpoint that I want to share with you.

I cannot find any sample size calculation strategy in your methods, so it seems that it is a report of a challenging experience conducted on a small sample size of children. Moreover, you have not mentioned any predefined inclusion and exclusion criteria, which is necessary. When it comes to analysis, considering the very low sample size, discussing the results of the performed analyzes is very difficult and it may not be fundamentally correct to refer to analytical analyses.

In view of the above, I highly recommend you to re-write the whole paper as a brief report including descriptive results, without any analytical analysis. I also suggest you to summarize the introduction part in just 2-3 paragraphs including background, gap of knowledge, and your study logic.

Regards,

Author Response

The authors appreciate the time you devoted to reading our manuscript and helping us to craft an improved version of the investigation. We are pleased to clarify your concerns, which we believe have improved the quality and applicability of this work. Please, find below our responses to each of your observations. We have made a concerted attempt to systematically address the specific concerns raised for this revision and we have highlighted the alterations to this revision within the manuscript with track changes for your convenience.

Reviewer reports:

Reviewer 1:

Background

REVIEWER 1: I received your submission as a reviewer and have read it several times with great interest. I appreciate your great job which was so exciting in terms of conduction. However, there are several important concerns from my viewpoint that I want to share with you.

AUTHORS: Thank you for your comment. We will try to attend all your suggestions following your concerns.

REVIEWER 1: I cannot find any sample size calculation strategy in your methods, so it seems that it is a report of a challenging experience conducted on a small sample size of children.

AUTHORS: Thank you for your comment. Correct, this research was conducted in an experience we carried out in a local school with a small sample size of children; therefore, not a sample size calculation method was used, as the sample was composed by the total quantity of students from the class.

REVIEWER 1:  Moreover, you have not mentioned any predefined inclusion and exclusion criteria, which is necessary.

AUTHORS: Thank you for your comment. The unique inclusion criteria taken into account was the fact that the student must be enrolled in the class were the experience was carried out. We included the following sentence in the manuscript:

Lines: 248, 249. “and the unique inclusion criteria taken into account was the fact that the student must be enrolled in the class where the experience was carried out”

REVIEWER 1: When it comes to analysis, considering the very low sample size, discussing the results of the performed analyzes is very difficult and it may not be fundamentally correct to refer to analytical analyses.

AUTHORS: Thank you for your comment. The discussion of the text has been modified attempting to contextualizing the results more that extrapolating them.

For example:

Lines:341,342 “These results revealed that the students that participated in the intervention improved their first aid knowledge and skills…”.

REVIEWER 1: In view of the above, I highly recommend you to re-write the whole paper as a brief report including descriptive results, without any analytical analysis.

AUTHORS: Thank you for your comment. As you suggested we re-wrote the whole paper including descriptive reports, reorganizing it and avoiding any analytical analysis.

REVIEWER 1: I also suggest you to summarize the introduction part in just 2-3 paragraphs including background, gap of knowledge, and your study logic.

AUTHORS: Thank you for your comment. As you suggested we summarized the introduction section including background, gap of knowledge and the study logic.

Reviewer 2 Report

Thank you for the opportunity to review the article. The work is interesting from a practical point of view, however, it needs several changes in the text.

1. I would like to ask you to clearly define the purpose of the work. It is currently formulated too broadly. The purpose should accurately reflect the author's intentions and relate to the results and methods used in the study

2 The introduction is disproportionately long in relation to the rest of the paper. The introduction should be a justification of the need for the study and some elements should be moved to the discussion

3) A detailed description of the study group, the place where the study was conducted and the study methodology should be considered.

4 The discussion should be significantly expanded

5. the Conclusions section is missing clearly

6. the summary should include more elements, especially regarding the results

7) Table 3 Questions related to Activity 2 -> Please consider describing the tables more precisely to make it more readable (Instead of Activity 2)

8. all abbreviations used in figures and tables must be described underneath them

9. there is a lack of information about the approval of the bioethics committee and parental/caregiver formal consent for the participation of their children in the study

10. there is a lack of information about the blinding of data and how they are processed.

Please pay attention to medical nomenclature

Author Response

The authors appreciate the time you devoted to reading our manuscript and helping us to craft an improved version of the investigation. We are pleased to clarify your concerns, which we believe have improved the quality and applicability of this work. Please, find below our responses to each of your observations. We have made a concerted attempt to systematically address the specific concerns raised for this revision and we have highlighted the alterations to this revision within the manuscript with track changes for your convenience.

Reviewer reports:

Reviewer 2:

REVIEWER 2: Thank you for the opportunity to review the article. The work is interesting from a practical point of view, however, it needs several changes in the text.

AUTHORS: Thank you for your comment, we will try to respond to your suggestions.

REVIEWER 2: 1. I would like to ask you to clearly define the purpose of the work. It is currently formulated too broadly. The purpose should accurately reflect the author's intentions and relate to the results and methods used in the study

AUTHORS: Thank you for your comment. As you suggested we tried to define the main objective of this research more clearly.

Lines: 248-250. “in order to see how the planned three activities would influence on the analyzed three years old kindergarten pupils”

REVIEWER 2: 2 The introduction is disproportionately long in relation to the rest of the paper. The introduction should be a justification of the need for the study and some elements should be moved to the discussion

AUTHORS: Thank you for your comment. As you suggested we summarized the introduction section,  giving a justification and background to our research, and we moved some paragraphs into discussion section, making this way the article more coherent and balanced.

REVIEWER 2: 3) A detailed description of the study group, the place where the study was conducted

AUTHORS: Thank you for your comment. As you suggested we added a more detailed description in order to provide a more specific description of the research context.

Lines: 251-254. “located in a small town with almost four thousand inhabitants. Taking this into account, this school is the only one in the village and its educational offer is from two years old to twelve years old, i.e. infant and primary education. The research was carried out”

REVIEWER 2: and the study methodology should be considered.

AUTHORS: Thank you for your comment. First, an ad-hoc instrument was designed based on the authors listed in the methodology section. This instrument was applied before and after the intervention. At each stage, the instrument was used by the researcher and the classroom teacher to collect information from the students through systematic observation. Subsequently, at the end of the intervention, means and standard deviations were calculated for each item and for the total; and pre and post scores were compared using Student's t-test for independent samples. Finally, to find out the effect size, Cohen's d index was calculated for each item as well as for the total.

We believe that this section is already well developed, however, if you consider that we should add some specific information please let us know and we will add it gladly.

REVIEWER 2: 4 The discussion should be significantly expanded

AUTHORS: Thank you for your comment. As you suggested, the discussion section was expanded, comparing our findings with previous investigations and discussing and interpreting them.

REVIEWER 2: 5. the Conclusions section is missing clearly

AUTHORS: Thank you for your comment. As you suggested we included a conclusion section.

Lines: 407-421

“Based on our findings, the three years old pupils group taking part in this research achieved a statistically significant improvement in the assessed first aid related knowledge and skills. Our finding could be relevant in the education system and from a public health perspective. Similarly, this research also shows the need for initial teacher training programs for preschool educators to empower them with the basic knowledge required by their future students (4). This would enable them to carry out effective low-cost interventions as well. Likewise, this data is relevant to lawmakers and politicians as it highlights the importance of designing a robust first aid curriculum for the preschool stage. Therefore, it can be said that first aid training should begin at this pre-school stage (31), thus forming, and ensuring knowledge and skills that will advocate for public health. By formally evaluating first aid training, we can establish guidelines for training and intervention methods and first aid knowledge in school children. This, in turn, will contribute to strengthening the community's ability to provide effective first aid”

REVIEWER 2: 6. the summary should include more elements, especially regarding the results

AUTHORS: Thank you for your comment. As you suggested we specified better the result items or contents.

Line 16: “related to first aid general knowledge, first aid kit and CPR maneuver”

REVIEWER 2: 7) Table 3 Questions related to Activity 2 -> Please consider describing the tables more precisely to make it more readable (Instead of Activity 2)

AUTHORS: Thank you for your comment. As you suggested we described the 3 tables more precisely in other them to be better understood.

Line: 329. “first aid general knowledge”

Line: 336 “the items that first aid kit should include and the use of each item,”

Line 340 “PCR maneuver knowledge and about vital signs related,”

REVIEWER 2: 8. all abbreviations used in figures and tables must be described underneath them

AUTHORS: Thank you for your comment.

Table 1, Line 276

“CPR: Cardiopulmonary resuscitation  P.N.A: Protect, Notify, Assist”

Table 2, Line 325

P.N.A: Protect, Notify, Assist

Table 4, Line 336.

“CPR: Cardiopulmonary resuscitation”

REVIEWER 2: 9. there is a lack of information about the approval of the bioethics committee and parental/caregiver formal consent for the participation of their children in the study

AUTHORS: Thank you for your comment. The data was collected within a bachelor's final project, at the University of the Basque Country, and all participants signed the written consent, however, an ethics committee is not required in that academic context.

Other than that information about the informed consent can be found in this lines 280,281. “During this process, informed consent was obtained from both the school's management team and the students' families”

REVIEWER 2: 10. there is a lack of information about the blinding of data and how they are processed.

AUTHORS: Thank you for your comment. We consider that this information has been added in the procedure section. Lines: 277-287

The evaluated experience was carried out because one of the authors conducted a research visit to the selected center. In order to link the data from the pre-phase to the post-phase data, the class list number was used, thus anonymizing the obtained responses for the rest of the research team members. Apart from the class code, no other potentially confidential data such as place of residence, parents' phone numbers, or ID numbers were used. The data collected with the assessment instrument were anonymous, as they were not shared with the educational institution or in this study as individual details, but rather analyzed in a global manner. These data remained privately stored in a repository accessible only to the research team members and will be digitally deleted once the project's lifespan comes to an end.

However, if you think that some of this information should be added in the article, please let us know and we will gladly add it.

Reviewer 3 Report

 Reconsider after major revision

Title:

Effectiveness of an Intervention to enhance first aid knowledge 2 among early childhood education students

The reviewer’s comments

The subject matter of this theme is good and well worth pursuing. This manuscript is well written, and a worthy contribution to the effectiveness of an intervention to enhance first aid knowledge among early childhood education students. However, the reviewer would like to see some revisions made to your manuscript.

1.     In the section of the introduction, I suggest the author(s) should provide what new

insight is your study offering to readers?

2.     In the section of the discussion, I suggest the author should strengthen more theoritical literatures to dialogue with the results.

3.     Please strengthen the conclusion and implications. Good finding suggestions for future practitioners and researchers. Authors are suggested for deeper

reflection, conclusions, and future recommendations by the reviewer.

4.     This study is interesting and innovative. Review once again after major revision.

Author Response

The authors appreciate the time you devoted to reading our manuscript and helping us to craft an improved version of the investigation. We are pleased to clarify your concerns, which we believe have improved the quality and applicability of this work. Please, find below our responses to each of your observations. We have made a concerted attempt to systematically address the specific concerns raised for this revision and we have highlighted the alterations to this revision within the manuscript with track changes for your convenience.

Reviewer reports:

Reviewer 3:

REVIEWER 3: The subject matter of this theme is good and well worth pursuing. This manuscript is well written, and a worthy contribution to the effectiveness of an intervention to enhance first aid knowledge among early childhood education students. However, the reviewer would like to see some revisions made to your manuscript.

AUTHORS: Thank you for your comment. We followed all your suggestions in order to improve our research.

REVIEWER 3: 1.     In the section of the introduction, I suggest the author(s) should provide what new insight is your study offering to readers?

AUTHORS: Thank you for your comment. As you suggested  we addedthe new insight that our study is offering to readers. Lines: 239-241.

“and still bigger when it comes to the youngest pupils within kindergarten, since three years old pupils have barely been addressed”

REVIEWER 3: 2.     In the section of the discussion, I suggest the author should strengthen more theoritical literatures to dialogue with the results.

AUTHORS: Thank you for your comment. As you suggested we rebuilt the discussion section, adding more theoretical literature and dialogue with our results. Lines 338-370.

REVIEWER 3: 3.     Please strengthen the conclusion and implications. Good finding suggestions for future practitioners and researchers. Authors are suggested for deeper reflection, conclusions, and future recommendations by the reviewer.

AUTHORS: Thank you for your comment. As you suggested we added a conclusion section and reflections in order to improve the quality of the manuscript, as well as future investigations were specified at the final paragraph on the discussion section.

Round 2

Reviewer 1 Report

According to the replies, I have one more suggestion, which is related to the tiltle. I highly recommend to add "a pilot study" to the end of the title to make it more sensible for the audiance.

Author Response

The authors appreciate the time you devoted to reading our manuscript and helping us to craft an improved version of the investigation. We are pleased to clarify your concerns, which we believe have improved the quality and applicability of this work. Please, find below our responses to each of your observations. We have made a concerted attempt to systematically address the specific concerns raised for this revision and we have highlighted the alterations to this revision within the manuscript with track changes for your convenience.

Reviewer reports:

Reviewer 1:

Background

REVIEWER 1: According to the replies, I have one more suggestion, which is related to the tiltle. I highly recommend to add "a pilot study" to the end of the title to make it more sensible for the audiance.

AUTHORS: Thank you so much for your comment. As you suggested we added “A pilot Study” in Line 3.

Reviewer 2 Report

The authors addressed all the problems and errors in the reviews. I propose to accept the article for publication.

Author Response

The authors appreciate the time you devoted to reading our manuscript and helping us to craft an improved version of the investigation. We are pleased to clarify your concerns, which we believe have improved the quality and applicability of this work. Please, find below our responses to each of your observations. We have made a concerted attempt to systematically address the specific concerns raised for this revision and we have highlighted the alterations to this revision within the manuscript with track changes for your convenience.

Reviewer reports:

Reviewer 2:

REVIEWER 2: The authors addressed all the problems and errors in the reviews. I propose to accept the article for publication.

AUTHORS: Thank you so much for your revision and your current proposition to be published.

Reviewer 3 Report

Title:

Effectiveness of an Intervention to enhance first aid knowledge among early childhood education students

The reviewer’s comments

The subject matter of this theme is good and well worth pursuing. This manuscript is well written, and a worthy contribution to the erffectiveness of an intervention to enhance first aid knowledge among early childhood education students

However, the reviewer would like to see some revisions made to your manuscript before publication.

1)     In the section of the introduction, the research purposes can be listed.

2)     Please strengthen the theoretical perspective to explain the association and build arguments in your study.

3)     In the section of the discussion, I suggest the author should provide more theoritical literatures to dialogue with the results.

4)     Please strengthen the conclusion and implications. Good finding suggestions for future practitioners and researchers.

5)    This study is interesting and innovative. Review once again after major revision.

Title:

Effectiveness of an Intervention to enhance first aid knowledge among early childhood education students

The reviewer’s comments

The subject matter of this theme is good and well worth pursuing. This manuscript is well written, and a worthy contribution to the erffectiveness of an intervention to enhance first aid knowledge among early childhood education students

However, the reviewer would like to see some revisions made to your manuscript before publication.

1)     In the section of the introduction, the research purposes can be listed.

2)     Please strengthen the theoretical perspective to explain the association and build arguments in your study.

3)     In the section of the discussion, I suggest the author should provide more theoritical literatures to dialogue with the results.

4)     Please strengthen the conclusion and implications. Good finding suggestions for future practitioners and researchers.

5)    This study is interesting and innovative. Review once again after major revision.

Author Response

The authors appreciate the time you devoted to reading our manuscript and helping us to craft an improved version of the investigation. We are pleased to clarify your concerns, which we believe have improved the quality and applicability of this work. Please, find below our responses to each of your observations. We have made a concerted attempt to systematically address the specific concerns raised for this revision and we have highlighted the alterations to this revision within the manuscript with track changes for your convenience.

Reviewer reports:

Reviewer 3:

REVIEWER 3: The subject matter of this theme is good and well worth pursuing. This manuscript is well written, and a worthy contribution to the erffectiveness of an intervention to enhance first aid knowledge among early childhood education students

However, the reviewer would like to see some revisions made to your manuscript before publication.

AUTHORS: Thank you for your comment. We will try to attend all your suggestions.

REVIEWER 3: 1)     In the section of the introduction, the research purposes can be listed.

AUTHORS: Thank you for your comment. As you suggested we tried to make the research purpose clearer setting it at the end of the introduction section.

Lines: 141-143 “Therefore, the aim of this study was to see whether it is possible to increase first aid knowledge and skills among a sample of preschool students through a low-cost intervention.”

REVIEWER 3: 2)     Please strengthen the theoretical perspective to explain the association and build arguments in your study.

AUTHORS: Thank you for your comment. As you suggested we tried to strengthen the theoretical perspective and building new arguments during the article:

Lines 39-41: “Moreover, the interventions carried out at the school context overall differs among different investigations (10) not being thus clear which type of intervention is more effective (9)”

  1. “He Z, Wynn P, Kendrick D. Non-resuscitative first-aid training for children and laypeople: a systematic review. Emergency Medicine Journal. 2014 Sep;31(9):763–8.”

Lines 66,67: “being this one of the most used content in previous intervention research (9)”

Lines 129-131: Another first aid training investigation, carried out with 150 kindergarten children, showed that this type of training courses help preschool children to act and provide help in emergency situations (35)  .

  1. Mohajervatan A, Raeisi AR, Atighechian G, Tavakoli N, Muosavi H. The Efficacy of Operational First Aid Training Course in Preschool Children. Health in Emergencies & Disasters Quarterly. 2020 Oct 1;6(1):17–22.

Lines 131-134: “. In a recent pilot study carried out with kindergarten students, pupils knowledges and skills were significantly improved regarded to first aid, showing some preliminary results that indicated that first aid trainning may be beneficial for pre school students (7)”

REVIEWER 3: 3)     In the section of the discussion, I suggest the author should provide more theoritical literatures to dialogue with the results.

AUTHORS: Thank you for your comment. As you suggested we tried to improve the discussion section adding more dialogue and theoretical literature within the article.

Lines 266-268: “Among this investigation, the research carried out by Mohajervatan et al (35) deserves special attention, since it was carried out with a 150-kindergarten sample, and showed improvements in children’s first aid knowledge, being these results in consonance with our results in our investigation.”

  1. “He Z, Wynn P, Kendrick D. Non-resuscitative first-aid training for children and laypeople: a systematic review. Emergency Medicine Journal. 2014 Sep;31(9):763–8.”
  2. Mohajervatan A, Raeisi AR, Atighechian G, Tavakoli N, Muosavi H. The Efficacy of Operational First Aid Training Course in Preschool Children. Health in Emergencies & Disasters Quarterly. 2020 Oct 1;6(1):17–22.

Lines 286-290: “even though these cited investigations were carried out mostly with high school context, being many lees analyzed younger pupils. In this line, within the mixed contents that different first aid interventions show (9)  previous research, in line with our investigation,  show that the used resuscitative and non-resuscitative programs can provide kindergarten pupils with first aid training (7,32,35) being this an aspect that need to be considered.”

REVIEWER 3: 4)     Please strengthen the conclusion and implications. Good finding suggestions for future practitioners and researchers.

AUTHORS: Thank you for your comment. As you suggested we tried to strengthen the conclusion section and implications, as well as suggestions for future practitioners.

Lines 338,339: “showing a high effect size from pre-phase to the post-phase, filling thus a described limitation in the literature (9)”

Line 340: “When it comes to practitioners”

Lines 350-352: “Regarding to future research, the literature in this field shows a clear limitation, the analyzed groups are small (9) so researchers should work trying to fill this gap.”

REVIEWER 3: 5)    This study is interesting and innovative. Review once again after major revision.

AUTHORS: Thank you for your comment. We tried to address all your interesting suggestions and we hope you find the article improved.

Round 3

Reviewer 1 Report

Dear authors,

Thank your for the revisions, I vote for accept.

Kind Regards,

Reviewer 3 Report

Thanks to the author for the correction. Revisions or explanations are all made according to the suggestions of the reviewer. Accept in present form.

Thanks to the author for the correction. Revisions or explanations are all made according to the suggestions of the reviewer. Accept in present form.